# Associations between maternal characteristics and pharmaceutical treatment of gestational diabetes: an analysis of the UK Born in Bradford (BiB) cohort study

Gilberte Martine-Edith  ,[1] William Johnson,[1] Eugenie Hunsicker,[2] Mark Hamer,[3] Emily S Petherick[1]

[1]School of Sport, Exercise and Health Sciences, Loughborough University, Loughborough, UK
[2]School of Science, Loughborough University, Loughborough, UK
[3]Institute of Sport, Exercise and Health, Division Surgery Interventional Science, University College London, London, UK

**Correspondence to**
Dr William Johnson;
w.o.johnson@lboro.ac.uk

## ABSTRACT

**Objectives** To identify the maternal characteristics associated with pharmaceutical treatment of gestational diabetes mellitus (GDM).

**Design** Prospective birth cohort study.

**Setting** Bradford, UK.

**Participants** 762 women from the Born in Bradford (BiB) cohort who were treated for GDM in a singleton pregnancy. BiB cohort participants were recruited from 2007 to 2010. All women booked for delivery were screened for GDM between 26 and 28 weeks of gestation using a 75 g 2-hour oral glucose tolerance test (OGTT).

**Outcome measure** GDM treatment type: lifestyle changes advice (lifestyle changes), lifestyle changes advice with supplementary insulin (insulin) and lifestyle changes advice with supplementary metformin (metformin).

**Results** 244 (32%) women were prescribed lifestyle changes advice alone while 518 (68%) were offered supplemental pharmaceutical treatment. The odds of receiving pharmaceutical treatment relative to lifestyle changes advice alone were increased for mothers who were obese (OR 4.6, 95% CI 2.8 to 7.5), those who smoked (OR 2.6, 95% CI 1.2 to 5.5) and had higher fasting glucose levels at OGTT (OR 2.1, 95% CI 1.6 to 2.7). The odds of being prescribed pharmaceutical treatment rather than lifestyle changes advice were lower for Pakistani women (OR 0.7, 95% CI 0.4 to 1.0)) than White British women. Relative to insulin treatment, metformin was more likely to be offered to obese women than normal weight women (relative risk ratio, RRR 3.2, 95% CI 1.3 to 7.8) and less likely to be prescribed to women with higher fasting glucose concentrations at OGTT (RRR 0.3, 95% CI 0.2 to 0.6).

**Conclusions** In the BiB cohort, GDM pharmaceutical treatment tended to be prescribed to women who were obese, White British, who smoked and had more severe hyperglycaemia. The characteristics of metformin-treated mothers differed from those of insulin-treated mothers as they were more likely to be obese but had lower glucose concentrations at diagnosis.

### Strengths and limitations of this study

► This study was based on a large sample of women diagnosed with gestational diabetes mellitus (GDM) in a centre where universal GDM screening was in place.
► Data used for this study captured a key transitional period in GDM management as metformin was introduced as an additional pharmaceutical treatment option.
► The mainly bi-ethnic nature of the sample allowed for the exploration of ethnic differences in GDM treatment between Pakistani and White British women.
► The generalisability of the findings might be limited by the fact that this was a single-centre observational study.
► The number of women treated with supplemental metformin was relatively small compared with the two other treatment types.

## INTRODUCTION

Gestational diabetes mellitus (GDM) is one of the most common complications of pregnancy.[1] In 2019, the International Diabetes Federation estimated that 13.2% of pregnancies, or 17 million live births, were affected by GDM worldwide.[2] The reported prevalence of GDM is 5% in the UK.[3] Ethnicity is a risk factor for GDM and in particular, South Asian (SA) women have been shown to have a higher risk for GDM than White women.[4–6] The public health significance of GDM lies in the intergenerational cycle of diabetes and obesity risk it perpetuates as GDM is associated with both maternal complications (eg, pre-eclampsia, caesarean delivery) and health risks for the offspring (eg, macrosomia, childhood obesity).[7]

Guidelines for initial GDM management recommend lifestyle changes (dietary and exercise advice).[8 9] While these changes are largely effective, hyperglycaemia persists for 15%–30% of women and supplemental pharmacological treatment is required.[8] Historically, subcutaneous insulin was the first-line pharmacological agent.[8] However, metformin has been increasingly accepted following the Metformin in Gestational diabetes (MiG) trial that validated it as a safe alternative to insulin,[10] despite uncertainties regarding its long-term effects on offspring health.[11] In the UK, both the 2008 and 2015 National Institute for Health and Care Excellence guidelines initially recommend metformin for GDM treatment and insulin is suggested when metformin is contraindicated, not tolerated or ineffective.[9]

With the aim to inform clinical management of GDM, previous research has investigated the characteristics associated with the need for supplemental pharmacological treatment in mothers with GDM.[12–18] High maternal body mass index (BMI), history of GDM, advanced age and adverse oral glucose tolerance test (OGTT) were among factors increasing the probability of receiving pharmacological treatment compared with lifestyle changes advice alone. However, there is still limited evidence of the associations between maternal characteristics and GDM pharmaceutical treatment in the UK.[17–21] Also, despite the known differences in the risk of GDM between SA and White women, the differences in their risk for GDM pharmaceutical treatment relative to lifestyle changes advice remain largely under-researched.[13 22]

Using a largely bi-ethnic UK birth cohort that included women with GDM treated both before and after metformin introduction, this study aimed to identify the maternal characteristics associated with GDM pharmacological treatment.

## METHODS

### Study

Born in Bradford (BiB) is a longitudinal prospective birth cohort study.[23] Bradford, a city in the north of England, constitutes a multi-ethnic population of more than 500 000 individuals, with 20% of the population of SA origin. Data were collected between 2007 and 2010 from 12 453 women (and their partners and offspring) booked for delivery at the Bradford Royal Infirmary.[24]

### Patient and public involvement

This was a secondary analysis of data from the BiB cohort. BiB has a number of established community advisory groups who are involved in the design, conduct, reporting and dissemination of findings from the BiB research programme.

### Sample

Our sample comprised 762 women with data on maternal characteristics (figure 1). Cohort participants diagnosed with GDM in a singleton pregnancy were included if

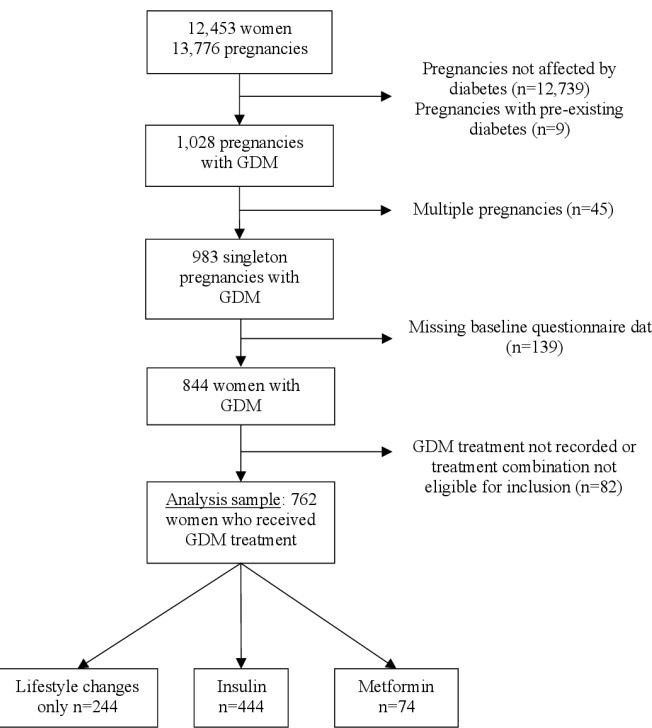

**Figure 1** Flow chart of study participation. GDM, gestational diabetes mellitus.

they received (1) lifestyle changes advice only, (2) lifestyle changes advice with supplementary insulin or (3) lifestyle changes advice supplemented by metformin. We excluded GDM treatment combinations (eg, lifestyle changes advice supplemented by both metformin and insulin treatment) that did not yield sufficient numbers for meaningful analyses to be conducted. Participants with GDM for whom treatment was not recorded were excluded. If mothers had more than one singleton pregnancy affected and treated for GDM during the study, we only included the first pregnancy. Singleton pregnancies not affected by GDM and higher order pregnancies (twins, triplets) whether or not affected by GDM were excluded from the study, as were women with pre-existing diabetes.

### Screening and diagnosis of GDM

All women enrolled in the BiB study were offered GDM screening. This was conducted between 26 and 28 weeks of gestation using the 2-hour 75 g OGTT and 80% of women attended their appointment.[24] Diagnosis of GDM was made using the modified 1999 WHO criteria in accordance with local recommendations at the time of recruitment (fasting glucose concentration ≥6.1 mmol/L and/or 2-hour postload glucose ≥7.8 mmol/L).[25]

### Management and treatment of GDM

Local procedure meant that all women were referred to the joint obstetric diabetes clinic following a diagnosis of GDM. Women were educated in dietary and exercise changes and capillary glucose monitoring. Individualised dietary recommendations were provided by a dietician

and daily walking for at least 30 min was recommended. If glucose targets were achieved after a week (fasting plasma glucose: 4.0–5.5 mmol/L; 2-hour postprandial: ≤7.5 mmol/L), lifestyle changes were continued without additional pharmacological treatment. If hyperglycaemia persisted, treatment was supplemented with insulin injections until delivery in the first part of the study (04/2007-03/2009). Following metformin introduction (04/2009), both insulin injections and metformin tablets (850 mg, two times per day) were pharmacological prescription options.

### Study outcome: GDM treatment type

The three reported treatment options evaluated in our study were: counselling for lifestyle changes, insulin and metformin. Lifestyle changes consisted of diet and exercise. Insulin and metformin groups included women who initially received lifestyle changes advice followed by supplementary insulin and metformin treatment, respectively.

### Maternal characteristics
#### Sociodemographic characteristics

Seven sociodemographic characteristics were considered: age at childbirth, marital and cohabitation status, ethnicity (White British, Pakistani, other), employment status (previously, currently or never employed), migration status, educational levels and parity. These were self-reported using interviewer-administered questionnaires at booking conducted in English or South Asian languages (eg, Bengali, Punjabi). Ethnicity was grouped according to the UK Office of National Statistics guidelines.[26] Education levels corresponded to ≤5 General Certificate of Secondary Education qualification, A level equivalent, higher than A level and other/unknown. Migration status was classified in two groups: mother was born in the UK or moved to the UK at ≤5 years old and mother moved to the UK >5 years of age. Marital and cohabitation status was defined as married and living with a partner, not married and living with a partner or not living with a partner.

#### Lifestyle and health characteristics

Nine lifestyle and health variables were analysed: BMI at booking, smoking during pregnancy (yes/no), physical activity levels, family history of diabetes (yes/no), history of GDM before the study (yes/no), pre-existing hypertension (yes/no), gestational age and blood glucose concentrations at OGTT (fasting and 2-hour postload) and start date of treatment relative to metformin introduction (before/after). Maternal BMI was obtained from height and weight measurements conducted at recruitment using Leicester Height Measure and Seca digital scales. Family history of diabetes, history of GDM and pre-existing hypertension were self-reported. Gestational age was recorded, and plasma glucose levels were measured at OGTT using a glucose oxidase method. Maternal physical activity levels (inactive, moderately inactive, moderately active, active) were self-reported using the UK General Practice Physical Activity Questionnaire.[27]

### Statistical analysis

Analyses were based on two time periods to account for the fact that metformin was used for GDM treatment in the study from April 2009 onwards, which is 2 years after the first women with GDM were offered lifestyle changes advice with or without insulin treatment in the cohort.

### Overall study period: April 2007–February 2011
#### Descriptive analysis

Using the whole study sample, we considered two treatment types: lifestyle changes advice and pharmaceutical treatment (ie, insulin-treated and metformin-treated women were grouped). Differences in maternal characteristics between women receiving lifestyle changes advice alone and those receiving supplemental pharmaceutical treatment were explored using the Mann-Whitney U test for continuous variables and $\chi^2$ (or Fisher's exact) test for categorical variables. The Holm-Bonferroni correction adjusted for multiple testing.[28 29]

#### Regression analysis

Variable selection for the binary logistic regression model was conducted using the least absolute shrinkage and selection operator (LASSO) which shrinks less stable coefficients exactly to zero, allowing for the selection of a more parsimonious model.[30] For each maternal characteristic selected through LASSO, a regression model was fitted to assess the unadjusted relationships between maternal characteristic and GDM pharmaceutical treatment, relative to lifestyle changes advice. The associations between maternal characteristics and GDM treatment were further assessed in a fully adjusted model, including all maternal characteristics.

#### Sensitivity analysis

Given the higher risk for insulin resistance and GDM in Pakistani women compared with White British women in the BiB cohort,[31] we reproduced the whole sample analysis but stratified by ethnicity, to evaluate whether the associations between maternal characteristics and GDM pharmaceutical treatment were influenced by ethnicity. Differences in maternal characteristics between White British and Pakistani women were also examined.

### Period after metformin introduction: April 2009–February 2011
#### Descriptive analysis

Using the subsample of women who started GDM treatment after metformin introduction, we considered three treatment types: lifestyle changes advice, insulin and metformin. The differences in the LASSO-selected maternal characteristics were examined by GDM treatment type. The Kruskal-Wallis test was used for continuous variables and the $\chi^2$ (or Fisher's exact) test was used for categorical variables. The Holm-Bonferroni correction adjusted for multiple testing.[28 29]

### Regression analysis

The relationships between maternal characteristics and insulin and metformin treatment were compared with lifestyle changes advice alone, in a multinomial logistic regression model including the LASSO-selected characteristics. The same multinomial logistic regression was fitted but using insulin as the reference group, to examine the maternal characteristics associated with metformin rather than insulin (the associations between maternal characteristics and lifestyle changes advice relative to insulin were omitted).

Analyses were conducted using R (R V.3.4.1 & R Studio V.1.0.153 for Windows) and Stata/SE software (Stata/SE V.15 for Windows; StataCorp).

### RESULTS

A total of 844 women were diagnosed with GDM in a singleton pregnancy. Eighty-two women who did not meet treatment inclusion criteria were excluded, leading to a sample of 762 women (figure 1).

### Overall study period: lifestyle changes vs pharmaceutical treatment

Thirty-two per cent of women received lifestyle changes advice alone and 68% received supplemental pharmacological treatment during the study (table 1). Women who were prescribed pharmacological treatment were older at childbirth (median age: 31.7 years (IQR, 7.6) compared with women receiving lifestyle changes advice (29.9 years (IQR, 8.1)), they were more hyperglycaemic at OGTT and had higher obesity rates (41.7% vs 19.0%). These differences remained statistically significant after accounting for multiple testing (table 1).

A total of 12 maternal characteristics were selected via LASSO and these were included in the regression analysis (table 2). Unadjusted analysis showed that obese women had five times the odds of being reported to have been offered pharmaceutical treatment (OR 4.6, 95% CI 2.8 to 7.5) than lifestyle changes advice. The odds of pharmaceutical treatment compared with lifestyle changes advice were higher for women who smoked during pregnancy (OR 2.6, 95% CI 1.2 to 5.5) and those who had higher fasting glucose concentrations at OGTT (OR 2.1, 95% CI 1.6 to 2.7). Relative to White British women, Pakistani women were predicted to have lower odds of being prescribed pharmaceutical treatment (OR 0.7, 95% CI 0.4 to 1.0) (table 2). Obesity and smoking were less prevalent among Pakistani women than White British women (online supplemental table 1).

Fully adjusted analyses confirmed that obesity, smoking and higher glucose concentrations at diagnosis were associated with higher odds of pharmaceutical treatment although the estimates were attenuated (table 2). Adjusting for fasting glucose weakened the relationships between obesity and pharmaceutical treatment. Adjustments for ethnicity brought the estimates for smoking closer towards the null.

The sensitivity analysis showed that for both White British and Pakistani women, higher glucose concentrations at OGTT and obesity were associated with an increase in the odds of being prescribed pharmaceutical treatment relative to lifestyle changes advice alone (online supplemental table 2).

### Period after metformin introduction: lifestyle changes advice versus insulin versus metformin

After metformin introduction, 31.1% of women received lifestyle changes advice alone, 50.5% were prescribed supplemental insulin and 18.4% were offered supplemental metformin (table 3). Mothers in the lifestyle changes group were more likely to be younger, less hyperglycaemic and have a lower BMI than women receiving supplemental insulin or metformin.

Relative to lifestyle changes advice, the risk of insulin treatment was 2.3 times higher for both obese women and women with higher fasting glucose concentrations at OGTT (table 4). The risk of insulin treatment relative to lifestyle changes advice was also higher for women who smoked during pregnancy compared with those who did not smoke. Supplemental metformin treatment rather than lifestyle changes advice alone was 7.3 times (95% CI 2.7 to 20.0) more likely for obese women.

Compared with insulin treatment, the risk of metformin was three times higher for obese than normal weight women and Pakistani women than White British women (table 4). Higher fasting glucose concentrations at OGTT were associated with a lower risk (relative risk ratio 0.3 (95% CI 0.2 to 0.6)) of a record of receiving metformin treatment relative to insulin.

### DISCUSSION

Our study showed that obesity, smoking and higher glucose concentrations at OGTT were key maternal characteristics associated with supplemental pharmaceutical treatment compared with lifestyle changes advice alone. Ethnic differences were also identified as, relative to White British women, Pakistani women were less likely to receive pharmaceutical treatment as a whole than lifestyle changes advice. Among women who received pharmaceutical treatment, metformin was more likely to be prescribed to obese women than normal weight women and to Pakistani women than White British women. Women who were more hyperglycaemic at diagnosis were more likely to be prescribed insulin rather than metformin.

Lifestyle changes advice supplemented by pharmaceutical treatment was the most common form of GDM management in our study. This contrasted with previous studies in which mothers with GDM were more frequently managed with lifestyle changes advice.[12] [14–17] [32–34] These disparities could be due to differences in GDM diagnostic criteria: the modified 1999 WHO criteria in our study used higher fasting glucose thresholds at OGTT but lower 2-hour thresholds than other criteria in by previous

**Table 1**  Maternal characteristics by GDM treatment type across the whole study period (2007–2011)

| | Lifestyle changes advice (n=244) | Pharmaceutical treatment (n=518) | P value | P value* | n (%) missing |
|---|---|---|---|---|---|
| Start date of treatment†, n (%) | | | 0.486 | >0.999 | 4 (0.5) |
| Before metformin introduction (2007–2009) | 121 (49.6) | 241 (47) | | | |
| After metformin introduction (2009–2011) | 123 (50.4) | 273 (53) | | | |
| Age at childbirth‡ (years), median (IQR) | 29.9 (8.1) | 31.7 (7.6) | <0.001 | 0.001 | 0 |
| BMI at booking‡ (kg/m$^2$), median (IQR) | 25.2 (6.0) | 28.4 (7.9) | <0.001 | <0.001 | 46 (6.0) |
| BMI category at booking†, n (%) | | | <0.001 | <0.001 | 46 (6.0) |
| Underweight (BMI <18.5 kg/m$^2$) | 7 (3.0) | 7 (1.4) | | | |
| Normal weight (18.5≤BMI≤24.9 kg/m$^2$) | 107 (46.1) | 122 (25.2) | | | |
| Overweight (25.0≤BMI≤29.9 kg/m$^2$) | 74 (31.9) | 153 (31.6) | | | |
| Obese (BMI ≥30.0 kg/m$^2$) | 44 (19.0) | 202 (41.7) | | | |
| Smoking during pregnancy†, n (%) | | | 0.008 | 0.096 | 1 (0.1) |
| Yes | 11 (4.5) | 53 (10.2) | | | |
| No | 233 (95.5) | 464 (89.8) | | | |
| Parity†, n (%) | | | 0.671 | >0.999 | 24 (3.1) |
| 0 | 88 (37.0) | 164 (32.8) | | | |
| 1 | 53 (22.3) | 119 (23.8) | | | |
| 2 | 41 (17.2) | 99 (19.8) | | | |
| 3+ | 56 (23.5) | 118 (23.6) | | | |
| Physical activity levels†, n (%) | | | 0.684 | >0.999 | 109 (14) |
| Inactive | 134 (62.3) | 285 (65.1) | | | |
| Moderately inactive | 37 (17.2) | 77 (17.6) | | | |
| Moderately active | 35 (16.3) | 56 (12.8) | | | |
| Active | 9 (4.2) | 20 (4.6) | | | |
| Ethnic group†, n (%) | | | 0.06 | 0.54 | 0 |
| White British | 47 (19.3) | 140 (27.0) | | | |
| Pakistani | 152 (62.3) | 298 (57.5) | | | |
| Other | 45 (18.4) | 80 (15.4) | | | |
| Migration status†, n (%) | | | 0.253 | >0.999 | 13 (1.7) |
| Born in the UK or moved ≤5 years | 121 (51.3) | 286 (55.7) | | | |
| Moved to the UK >5 years | 115 (48.7) | 227 (44.2) | | | |
| Marital and cohabitation status†, n (%) | | | 0.239 | >0.999 | 0 |
| Married and living with partner | 198 (81.1) | 411 (79.3) | | | |
| Not married and living with partner | 21 (8.6) | 64 (12.4) | | | |
| Not living with partner | 25 (10.2) | 43 (8.3) | | | |
| Highest educational qualification†, n (%) | | | 0.528 | >0.999 | 4 (0.5) |
| 5 GCSE equivalent or less | 124 (51.0) | 273 (53.0) | | | |
| A-level equivalent | 28 (11.5) | 57 (11.1) | | | |
| Higher than A-level | 76 (31.3) | 141 (27.4) | | | |
| Other/unknown | 15 (6.2) | 44 (8.5) | | | |
| Family history of diabetes†, n (%) | | | 0.01 | 0.1 | 63 (8.3) |
| Yes | 128 (57.7) | 323 (67.7) | | | |
| No | 94 (42.3) | 154 (32.3) | | | |
| Pre-existing hypertension§, n (%) | | | | | 53 (7.0) |
| Yes | 3 (1.3) | 10 (2.1) | 0.563 | >0.999 | |
| No | 228 (98.7) | 468 (97.9) | | | |

Continued

**Table 1** Continued

| | Lifestyle changes advice (n=244) | Pharmaceutical treatment (n=518) | P value | P value* | n (%) missing |
|---|---|---|---|---|---|
| History of GDM before the study†, n (%) | | | 0.075 | 0.600 | 93 (12) |
| Yes | 10 (4.6) | 38 (8.4) | | | |
| No | 207 (95.4) | 414 (91.6) | | | |
| Mother's employment status†, n (%) | | | 0.008 | 0.096 | 0 |
| Currently employed | 90 (36.9) | 203 (39.2) | | | |
| Previously employed | 55 (22.5) | 159 (30.7) | | | |
| Never employed | 99 (40.6) | 156 (30.1) | | | |
| Gestational age at OGTT‡ (weeks), median (IQR) | 26.4 (1.6) | 26.3 (0.8) | 0.006 | 0.078 | 13 (1.7) |
| Fasting glucose concentrations at OGTT‡ (mmol/L), median (IQR) | 4.7 (0.7) | 5.1 (1.1) | <0.001 | <0.001 | 13 (1.7) |
| 2-hour post-load glucose concentrations at OGTT‡ (mmol/L), median (IQR) | 8.2 (0.8) | 8.6 (1.6) | <0.001 | <0.001 | 13 (1.7) |

Continuous data presented as median and IQR.
Categorical data presented as frequencies and percentages.
*Adjusted p value after Holm-Bonferroni correction.
†$\chi^2$ test.
‡Mann-Whitney U test.
§Fisher's exact test.
A-level, UK highest qualification in high school; BMI, body mass index; GCSE, general certificate of secondary education; GDM, gestational diabetes mellitus.

studies.[12 16 34] Additionally, the higher rates of pharmaceutical treatment in our study could reflect the higher risk profile of the BiB population and also, the high levels of deprivation in Bradford[24] could have limited health literacy and the adherence to lifestyle changes advice.[35]

Obesity, smoking during pregnancy and glucose concentrations at OGTT were the maternal characteristics most strongly associated with GDM supplemental pharmaceutical treatment in comparison to lifestyle changes advice alone. Previous research has also reported BMI as a risk factor for GDM pharmaceutical treatment, notably insulin. Although the specificity of these studies largely varied (eg, location, sample size, screening methods, diagnostic thresholds), they consistently showed that as maternal BMI increased, so did the risk of being treated with insulin.[12–15 32 34 36–38] Regarding the associations between smoking and GDM treatment, some studies showed that more smokers were treated with insulin than lifestyle changes,[15 33 39] while others found an opposite relationship,[38 40] although the differences between groups in these studies were not reported to be statistically significant. A more recent study has reported that smoking was associated with a higher risk of insulin treatment, although this was relative to women without GDM and women with GDM not requiring insulin treatment combined in the same control group.[41]

We hypothesise that the mechanisms explaining the associations between obesity, smoking, glucose concentrations at OGTT and GDM pharmaceutical treatment in our sample are closely related to obesity-induced and smoking-induced insulin resistance. Obesity may alter the functioning of pancreatic β-cells and exacerbates insulin resistance (which is already increased as a result of pregnancy).[22 33 36 38 42 43] Smoking has also been associated with insulin resistance, via processes including hormonal secretions (eg, growth hormone) that counteract insulin action.[44 45] Thus, although there was no direct measure of insulin resistance in this study, it is possible that women who were obese or smoked during pregnancy had a higher degree of insulin resistance. Additionally, in line with other studies,[16 34 46 47] we found that women who were prescribed pharmaceutical treatment were more likely to be more severely hyperglycaemic compared with women who received lifestyle changes advice alone. As increases in insulin resistance and β-cell dysfunction can further lead to higher glucose concentrations at the OGTT,[48–50] the severity of insulin resistance and its associated greater severity of hyperglycaemia in obese women and those who smoked could have been such that lifestyle changes advice alone were insufficient to achieve glucose targets. In that sense, our results accurately reflect clinical practice in Bradford as the decision to prescribe pharmaceutical treatment was based on the finding of glucose levels higher than the glucose targets. Further, what our study suggests is that the severity of hyperglycaemia may mediate the relationships between maternal obesity and smoking and GDM pharmaceutical treatment. This was confirmed by individual adjustment for fasting glucose which attenuated the relationships between obesity and GDM pharmaceutical treatment. This attenuation was however less evident for the relationships between

**Table 2** Associations between maternal characteristics and pharmaceutical treatment of GDM relative to lifestyle changes advice

| | Pharmaceutical treatment (n=372) | | | |
|---|---|---|---|---|
| | Unadjusted OR (95% CI) | P value | Adjusted OR (95% CI) | P value |
| Mother age at childbirth (years) | 1.1 (1.0 to 1.1) | <0.001 | 1.1 (1.0 to 1.1) | <0.001 |
| BMI categories at booking (kg/m$^2$) | | | | |
| Normal weight | Reference | | Reference | |
| Underweight | 0.8 (0.2 to 2.4) | 0.663 | 1.2 (0.4 to 4.3) | 0.725 |
| Overweight | 1.8 (1.1 to 2.7) | 0.008 | 1.3 (0.8 to 2.0) | 0.353 |
| Obese | 4.6 (2.8 to 7.5) | <0.001 | 3.0 (1.7 to 5.2) | <0.001 |
| Parity | | | | |
| 0 | Reference | | Reference | |
| 1 | 1.2 (0.7 to 1.9) | 0.475 | 0.6 (0.4 to 1.1) | 0.142 |
| 2 | 1.1 (0.7 to 1.9) | 0.588 | 0.6 (0.3 to 1.1) | 0.096 |
| 3+ | 1.3 (0.8 to 2.2) | 0.225 | 0.4 (0.2 to 0.9) | 0.022 |
| Ethnic origin | | | | |
| White British | Reference | | Reference | |
| Pakistani | 0.7 (0.4 to 1.0) | 0.081 | 0.6 (0.3 to 1.2) | 0.135 |
| Other | 0.5 (0.3 to 0.9) | 0.02 | 0.4 (0.2 to 0.8) | 0.015 |
| Highest educational qualification | | | | |
| 5 GCSE equivalent or less | Reference | | Reference | |
| A-level equivalent | 0.8 (0.5 to 1.5) | 0.554 | 0.7 (0.3 to 1.3) | 0.25 |
| Higher than A-level | 0.8 (0.5 to 1.2) | 0.219 | 0.7 (0.4 to 1.2) | 0.171 |
| Other/unknown | 1.0 (0.5 to 2.0) | 0.996 | 0.7 (0.3 to 1.6) | 0.396 |
| Employment status | | | | |
| Currently employed | Reference | | Reference | |
| Previously employed | 1.4 (0.9 to 2.2) | 0.161 | 1.1 (0.6 to 2.1) | 0.649 |
| Never employed | 0.7 (0.5 to 1.1) | 0.139 | 0.7 (0.4 to 1.3) | 0.244 |
| Physical activity levels | | | | |
| Active | Reference | | Reference | |
| Moderately active | 0.7 (0.3 to 1.9) | 0.538 | 0.7 (0.2 to 1.9) | 0.467 |
| Moderately inactive | 1.1 (0.4 to 2.8) | 0.793 | 1.1 (0.4 to 3.0) | 0.905 |
| Inactive | 1.0 (0.4 to 2.4) | 0.919 | 1.1 (0.4 to 2.9) | 0.882 |
| Smoking during pregnancy | 2.6 (1.2 to 5.5) | 0.011 | 1.9 (0.8 to 4.5) | 0.14 |
| Family history of diabetes | 1.3 (0.9 to 1.9) | 0.156 | 1.2 (0.8 to 1.9) | 0.337 |
| Gestational age at OGTT (weeks) | 0.9 (0.8 to 0.9) | 0.004 | 0.9 (0.8 to 1.0) | 0.045 |
| Fasting glucose at OGTT (mmol/L) | 2.1 (1.6 to 2.7) | <0.001 | 1.7 (1.3 to 2.3) | <0.001 |
| 2-hour post-load glucose at OGTT (mmol/L) | 1.5 (1.2 to 1.7) | <0.001 | 1.4 (1.1 to 1.7) | <0.001 |

A-level, UK highest qualification in high school; BMI, body mass index; GCSE, General Certificate of Secondary Education; GDM, gestational diabetes mellitus; OGTT, oral glucose tolerance test.

smoking and GDM pharmaceutical treatment, possibly due to the low proportion of smokers.

Another important finding of this study is that, relative to White British women, Pakistani women were predicted to have a lower risk for pharmaceutical treatment (when insulin and metformin treatment were grouped) compared with lifestyle changes alone. This may seem counterintuitive given SA women are more prone to insulin resistance than White European women due to a greater susceptibility to store adipose tissue viscerally rather than subcutaneously.[6 51] Wong[22] and Wong and Jalaludin[13] have also described that SA women had a lower risk to be prescribed with supplemental insulin rather than lifestyle changes advice alone than Anglo-Europeans. The authors suggested that this may be due to differences between the two ethnic groups

**Table 3** Maternal characteristics by GDM treatment type after metformin introduction (2009–2011)

| | Lifestyle changes advice (n=123) | Insulin (n=200) | Metformin (n=73) | P value | P value* | n (%) missing |
|---|---|---|---|---|---|---|
| Age at childbirth† (years), median (IQR) | 29.1 (7.8) | 31.8 (8.2) | 30.6 (9.0) | <0.001 | 0.004 | 0 |
| BMI at booking† (kg/m$^2$), median (IQR) | 24.7 (4.9) | 28.1 (9.2) | 29.3 (6.5) | <0.001 | 0.001 | 21 (2.8) |
| BMI category at booking‡ n (%) | | | | <0.001 | <0.001 | 21 (2.8) |
| Underweight (BMI <18.5 kg/m$^2$) | 4 (3.4) | 5 (2.7) | 0 | | | |
| Normal weight (18.5≤BMI≤24.9 kg/m$^2$) | 57 (48.3) | 55 (29.3) | 16 (23.2) | | | |
| Overweight (25.0≤BMI≤29.9 kg/m$^2$) | 40 (33.9) | 51 (27.1) | 22 (31.9) | | | |
| Obese (BMI ≥30.0 kg/m$^2$) | 17 (14.4) | 77 (41.0) | 31 (44.9) | | | |
| Smoking during pregnancy§, n (%) | | | | 0.004 | 0.032 | 0 |
| Yes | 4 (3.3) | 29 (14.5) | 6 (8.2) | | | |
| No | 119 (96.7) | 171 (85.5) | 67 (91.8) | | | |
| Parity§ n (%) | | | | 0.145 | 0.58 | 6 (0.8) |
| 0 | 52 (43.0) | 68 (34.5) | 22 (30.6) | | | |
| 1 | 26 (21.5) | 46 (23.3) | 21 (29.2) | | | |
| 2 | 18 (14.9) | 35 (17.8) | 19 (26.4) | | | |
| 3+ | 25 (20.7) | 48 (24.4) | 10 (13.9) | | | |
| Physical activity levels§ n (%) | | | | 0.424 | 0.63 | 1 (0.1) |
| Inactive | 71 (57.7) | 116 (58.3) | 43 (58.9) | | | |
| Moderately inactive | 23 (18.7) | 43 (21.6) | 9 (12.3) | | | |
| Moderately active | 22 (17.9) | 30 (15.1) | 13 (17.8) | | | |
| Active | 7 (5.7) | 10 (5.0) | 8 (11.0) | | | |
| Ethnic group§ n (%) | | | | 0.21 | 0.63 | 0 |
| White British | 24 (19.5) | 54 (27.0) | 15 (20.5) | | | |
| Pakistani | 74 (60.2) | 113 (56.5) | 50 (68.5) | | | |
| Other | 25 (20.3) | 33 (16.5) | 8 (11.0) | | | |
| Highest educational qualification§ n (%) | | | | 0.297 | 0.63 | 0 |
| 5 GCSE equivalent or less | 53 (43.1) | 104 (52.0) | 32 (43.8) | | | |
| A-level equivalent | 17 (13.8) | 27 (13.5) | 7 (9.6) | | | |
| Higher than A-level | 44 (35.8) | 55 (27.5) | 31 (42.5) | | | |
| Other/unknown | 9 (7.3) | 14 (7.0) | 3 (4.1) | | | |
| Family history of diabetes§ n (%) | | | | 0.099 | 0.553 | 27 (3.5) |
| Yes | 65 (57.0) | 130 (69.1) | 42 (62.7) | | | |
| No | 49 (43.0) | 58 (30.8) | 25 (37.3) | | | |
| Mother's employment status§ n (%) | | | | 0.079 | 0.553 | 0 |
| Currently employed | 48 (39.0) | 80 (40.0) | 28 (38.4) | | | |
| Previously employed | 19 (15.4) | 54 (27.0) | 20 (27.4) | | | |
| Never employed | 56 (45.5) | 66 (33.0) | 25 (34.2) | | | |
| Gestational age at OGTT† (weeks), median (IQR) | 26.3 (1.8) | 26.1 (0.8) | 26.3 (0.7) | 0.087 | 0.553 | 8 (1.0) |
| Fasting glucose concentrations at OGTT† (mmol/L), median (IQR) | 4.7 (0.8) | 5.2 (1.2) | 4.8 (0.7) | <0.001 | 0.001 | 8 (1.0) |
| 2-hour postload glucose concentrations at OGTT† (mmol/L), median (IQR) | 8.2 (0.8) | 8.6 (1.6) | 8.4 (1.3) | <0.001 | 0.001 | 8 (1.0) |

Continuous data presented as median and IQR.
Categorical data presented as frequencies and percentages.
*Adjusted p values after Holm-Bonferroni correction.
†Kruskal-Wallis test.
‡Fisher's exact test.
§$\chi^2$ test.
A-level, UK highest qualification in high school; BMI, body mass index; GCSE, general certificate of secondary education; GDM, gestational diabetes mellitus; OGTT, oral glucose tolerance test.

**Table 4** Associations between maternal characteristics and GDM pharmaceutical treatment after metformin introduction

| | Insulin and metformin treatment relative to lifestyle changes advice | | | | Metformin treatment relative to insulin | |
| | Insulin | | Metformin | | Metformin | |
| | Adjusted RRR (95% CI) | P value | Adjusted RRR (95% CI) | P value | Adjusted RRR (95% CI) | P value |
|---|---|---|---|---|---|---|
| Mother age at childbirth (years) | 1.1 (1.0 to 1.2) | 0.001 | 1.0 (1.0 to 1.1) | 0.16 | 0.9 (0.9 to 1.0) | 0.141 |
| BMI categories at booking (kg/m$^2$) | | | | | | |
| Normal weight | Reference | | Reference | | Reference | |
| Underweight | 1.8 (0.4 to 9.1) | 0.444 | – | – | – | – |
| Overweight | 0.6 (0.3 to 1.3) | 0.221 | 1.6 (0.6 to 4.0) | 0.29 | 2.6 (1.0 to 6.4) | 0.044 |
| Obese | 2.3 (1.0 to 5.2) | 0.051 | 7.3 (2.7 to 20.0) | <0.001 | 3.2 (1.3 to 7.8) | 0.01 |
| Parity | | | | | | |
| 0 | Reference | | Reference | | Reference | |
| 1 | 0.6 (0.3 to 1.4) | 0.234 | 0.8 (0.3 to 2.2) | 0.704 | 1.3 (0.5 to 3.3) | 0.524 |
| 2 | 0.5 (0.2 to 1.4) | 0.204 | 1.1 (0.4 to 3.2) | 0.817 | 2.1 (0.8 to 5.5) | 0.143 |
| 3+ | 0.6 (0.2 to 1.8) | 0.398 | 0.4 (0.1 to 1.6) | 0.21 | 0.7 (0.2 to 2.2) | 0.521 |
| Ethnic origin | | | | | | |
| White British | Reference | | Reference | | Reference | |
| Pakistani | 0.5 (0.2 to 1.4) | 0.197 | 1.7 (0.5 to 5.5) | 0.359 | 3.2 (1.1 to 9.3) | 0.031 |
| Other | 0.5 (0.2 to 1.4) | 0.187 | 0.7 (0.2 to 2.6) | 0.618 | 1.4 (0.4 to 4.7) | 0.605 |
| Highest educational qualification | | | | | | |
| 5 GCSE equivalent or less | Reference | | Reference | | Reference | |
| A-level equivalent | 0.6 (0.2 to 1.6) | 0.36 | 0.5 (0.1 to 1.9) | 0.325 | 0.8 (0.2 to 2.7) | 0.756 |
| Higher than A-level | 0.6 (0.3 to 1.4) | 0.271 | 1.0 (0.4 to 2.6) | 0.925 | 1.6 (0.7 to 3.7) | 0.281 |
| Other/unknown | 0.8 (0.2 to 2.4) | 0.645 | 0.4 (0.08 to 1.7) | 0.208 | 0.5 (0.1 to 2.1) | 0.328 |
| Employment status | | | | | | |
| Currently employed | Reference | | Reference | | Reference | |
| Previously employed | 1.6 (0.6 to 3.9) | 0.297 | 1.7 (0.6 to 5.0) | 0.306 | 1.1 (0.4 to 2.8) | 0.866 |
| Never employed | 0.8 (0.3 to 1.9) | 0.613 | 0.7 (0.2 to 2.3) | 0.617 | 0.9 (0.3 to 2.7) | 0.922 |
| Physical activity levels | | | | | | |
| Active | Reference | | Reference | | Reference | |
| Moderately active | 0.5 (0.1 to 2.0) | 0.374 | 0.4 (0.09 to 1.8) | 0.231 | 0.7 (0.2 to 2.9) | 0.647 |
| Moderately inactive | 1.0 (0.3 to 3.5) | 0.957 | 0.2 (0.05 to 1.1) | 0.063 | 0.2 (0.05 to 1.0) | 0.051 |
| Inactive | 0.8 (0.2 to 3.0) | 0.809 | 0.5 (0.1 to 2.1) | 0.328 | 0.6 (0.1 to 2.2) | 0.405 |
| Smoking during pregnancy | 4.1 (1.0 to 16.9) | 0.048 | 1.9 (0.3 to 11.7) | 0.474 | 0.5 (0.1 to 1.8) | 0.27 |
| Family history of diabetes | 1.2 (0.6 to 2.2) | 0.554 | 1.0 (0.5 to 2.1) | 0.959 | 0.8 (0.4 to 1.7) | 0.652 |
| Gestational age at OGTT (weeks) | 0.9 (0.8 to 1.1) | 0.364 | 1.0 (0.8 to 1.1) | 0.747 | 1.0 (0.9 to 1.2) | 0.644 |
| Fasting glucose at OGTT (mmol/L) | 2.3 (1.5 to 3.6) | <0.001 | 0.8 (0.4 to 1.4) | 0.456 | 0.3 (0.2 to 0.6) | <0.001 |
| 2-hour postload glucose at OGTT (mmol/L) | 1.2 (0.9 to 1.7) | 0.127 | 1.2 (0.9 to 1.8) | 0.245 | 1.0 (0.7 to 1.3) | 0.968 |

BMI, body mass index; GCSE, General Certificate of Secondary Education; GDM, gestational diabetes mellitus; OGTT, oral glucose tolerance test; RRR, relative risk ratio.

in the placental factors impacting, late in the pregnancy, on the severity of insulin resistance. Underlying ethnic differences in dietary habits during pregnancy and adherence to treatment could also be contributing factors.[22] In the context of our study, however, Pakistani mothers were less likely to report smoking and had lower BMI compared with White British women which was consistent with previous research in the BiB cohort.[24 52 53] This, combined with the fact that obesity and smoking were strongly associated with GDM pharmaceutical treatment in our study, may explain why Pakistani women were less likely to receive pharmaceutical treatment rather than lifestyle changes advice alone compared with White British women. The stratified analysis, however,

showed that higher maternal BMI and glucose concentration at OGTT were associated with a higher risk for pharmaceutical treatment relative to lifestyle changes advice, irrespective of maternal ethnicity.

The addition of metformin to the set of pharmacological options was not associated, at the time of the study, with any substantial shift in GDM management as insulin remained the most common pharmaceutical treatment prescribed. Nevertheless, we found that, obese women were more likely to be treated with metformin rather than insulin which is in line with a study by McGrath et al[54]. This perhaps is the result of clinical decision-making as metformin, compared with insulin, has been associated with lower weight gain[55] thus metformin could preferably be given to women with higher BMI. Further, we found that women with more severe hyperglycaemia were more likely to be prescribed insulin rather than metformin, which corroborated previous research.[54 56 57] As metformin is believed to act less rapidly than insulin,[18] it may be that in our study, even after metformin introduction, women with a higher severity of hyperglycaemia were preferentially prescribed insulin to promptly restore euglycaemia. Thus, it is somewhat surprising that Pakistani mothers, characteristically more hyperglycaemic and with lower BMI than White British women, were predicted to have a higher risk for metformin treatment compared with insulin than White British women. This may reflect individual treatment preference for metformin treatment as insulin injections are considered by mothers with GDM to be invasive and burdensome[58] and can be associated with social stigma within SA communities.[59] More research regarding the ethnic differences between metformin-treated and insulin-treated mothers with GDM would be needed to ascertain this finding.

The main strength of this study is that the findings are based on a large sample of women diagnosed with GDM from a cohort where universal GDM screening was in place. The data originated from a single diabetes clinic in the UK managed by the same senior clinician and where the same diagnostic criteria and glucose targets for GDM management were used throughout the study. This minimised bias related to differences in clinical practice and decisions between clinics. Another strength of our study is that, unlike previous studies that explored maternal characteristics of GDM treatment either before or after metformin introduction, our data captured GDM management both pre- and post-metformin introduction. This allowed for an analysis of the maternal characteristics associated with GDM pharmacological treatment during a key transitional period of changes in GDM management within the BiB cohort. Lastly, the mainly bi-ethnic nature of the BiB cohort enabled the assessment of the differences in the risk for GDM pharmaceutical treatment between Pakistani and White British mothers, which is particularly important given Pakistani mothers have a higher risk of developing GDM itself.

Our findings are, however, limited by the relatively small sample of women treated with metformin at the time of the BiB study compared with the other treatment types which means that our results must be interpreted with caution. We acknowledge that the generalisability of our results may be limited by the fact that this is a single-centre observational study, although our findings remained largely consistent with previous research.

To conclude, in the UK BiB cohort, women who received GDM supplemental pharmaceutical treatment rather than lifestyle changes advice alone were more likely to be obese, smokers, more hyperglycaemic and White British. Among women who received pharmaceutical treatment, the risk for metformin treatment was higher for Pakistani women and obese women, while women who were more hyperglycaemic were more likely to be prescribed insulin. Evaluation of the relationships between GDM treatment and maternal or offspring outcomes in the BiB cohort would thus have to account for the maternal determinants of GDM pharmaceutical treatment identified in this study.

**Acknowledgements** Born in Bradford is only possible because of the enthusiasm and commitment of the children and parents in BiB. We are grateful to all the participants, health professionals, schools and researchers who have made Born in Bradford happen. The authors would also like to thank Dr Donald Whitelaw for providing clinical insight regarding the management and treatment of GDM in the BiB cohort.

**Contributors** GME wrote the manuscript and was responsible for the acquisition, analysis and interpretation of data. EP and WJ revised the manuscript and contributed to the acquisition, analysis and interpretation of data. EH contributed to the analysis and interpretation of data and reviewed the manuscript. MH reviewed the manuscript. GME, WJ and EP are guarantors of this work and, as such, had full access to all the data in the study and take responsibility for the integrity of the data and the accuracy of the data analysis.

**Funding** This research was funded by Loughborough University and supported by the National Institute for Health Research (NIHR) Leicester Biomedical Research Centre. EP and WJ acknowledge support from the National Institute for Health Research (NIHR) Leicester Biomedical Research Centre, which is a partnership between University Hospitals of Leicester NHS Trust, Loughborough University, and the University of Leicester. WJ is supported by a UK Medical Research Council (MRC) New Investigator Research Grant (MR/P023347/1). Born in Bradford received funding from a Wellcome Trust infrastructure grant (WT101597MA), the National Institute for Health Research under its Collaboration for Applied Health Research and Care (CLAHRC) (IS-CLA-0113–10020). The NIHR Clinical Research Network which provided research delivery support for this study.

**Disclaimer** The views expressed in this paper are those of the authors and not necessarily those of the NIHR.

**Competing interests** None declared.

**Patient consent for publication** Not applicable.

**Ethics approval** Ethical approval for the study was granted by Bradford Research Ethics Committee (Ref 07/H1302/112).

**Provenance and peer review** Not commissioned; externally peer reviewed.

**Data availability statement** Scientists are encouraged and able to use BiB data. Data requests are made to the BiB executive using the form available from the study website http://www.borninbradford.nhs.uk. Guidance for researchers and collaborators, the study protocol and the data collection schedule are all available via the website. All requests are carefully considered and accepted where possible.

**ORCID iD**
Gilberte Martine-Edith http://orcid.org/0000-0002-4592-3680

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
