## [Reviewer comments · BMJ Open]

ARTICLE DETAILS

TITLE (PROVISIONAL)	Associations between maternal characteristics and pharmaceutical treatment of gestational diabetes: an analysis of the UK Born in Bradford (BiB) cohort study
AUTHORS	Martine-Edith, Gilberte; Johnson, William; Hunsicker, Eugenie; Hamer, Mark; Petherick, Emily

VERSION 1 – REVIEW

REVIEWER	Chaves, Catarina Serviço de Endocrinologia do Centro Hospitalar do Tâmega e Sousa
REVIEW RETURNED	28-Jun-2021

GENERAL COMMENTS	The manuscript is overall well-written and comprehensible, although there are some considerations I'd like to address. - In the abstract, it would be interesting to see a more detailed explanation of the cohort study, not so much about the statistical methods.- Please clarify why patients with missing data on ethnicity were excluded from the analysis.- As the pharmacological treatment is prescribed after the failure of dietary and exercise changes, the outcome in the study is in my opinion, failure in these lifestyle changes. If you look closely, some of the predictors you found are in fact, clinical findings that lead the doctor to prescribe a pharmacological treatment.
--

REVIEWER	Nagasubramanian, Vanitha Sri Ramachandra Institute of Higher Education and Research, Faculty of Pharmacy
REVIEW RETURNED	21-Jul-2021

GENERAL COMMENTS	The area of research kindles interest and the manuscript is well constructed. The title is broad and does not reflect on the association between maternal characteristics and Pharmacological treatment in women with GDM on Insulin and metformin which is the objective of the study. Title can be modified. Minor grammatical corrections need to be done in the abstract.
--

VERSION 1 – AUTHOR RESPONSE

Reviewer 1:

*In the abstract, it would be interesting to see a more detailed explanation of the cohort study, not so much about the statistical methods.

Response: The structure of the abstract has been modified to meet BMJ Open's guidelines and there is less emphasis on the statistical methods. Details regarding the period of data collection and method of GDM screening within the cohort have been added as follows: "BiB cohort participants were recruited from 2007 until 2010. All women booked for delivery were screened for GDM between 26 and 28 weeks of gestation using a 75g 2-hour glucose tolerance test (OGTT)" (p.2).

*Please clarify why patients with missing data on ethnicity were excluded from the analysis.

Response: The flowchart of study participation (Figure 1) has been corrected as we excluded women with no data on maternal baseline characteristics (available from a questionnaire completed at recruitment) rather than specifically data on ethnicity.

*As the pharmacological treatment is prescribed after the failure of dietary and exercise changes, the outcome in the study is in my opinion, failure in these lifestyle changes. If you look closely, some of the predictors you found are in fact, clinical findings that lead the doctor to prescribe a pharmacological treatment.

Response: It is stated in the manuscript that: "If hyperglycaemia persisted (following prescription of lifestyle changes advice), treatment was supplemented with insulin injections until delivery in the first part of the study (04/2007-03/2009). Following metformin introduction (04/2009), both insulin injections and metformin tablets (850 mg, twice daily) were pharmacological prescription options". Whilst our results may indeed reflect that lifestyle changes advice failed to restore euglycaemia, given the data available, it is impossible to determine whether 'lifestyle changes failure' is in fact a lack of patient adherence or lack of effectiveness of lifestyle changes. We therefore prefer to use a broader terminology and define the outcome of this study as 'GDM treatment type' (lifestyle changes advice, lifestyle changes advice with supplemental insulin and lifestyle changes with supplemental metformin).

Reviewer 2:

*The title is broad and does not reflect on the association between maternal characteristics and Pharmacological treatment in women with GDM on Insulin and metformin which is the objective of the study. Title can be modified.

Response: The title of the manuscript has been changed to: "Associations between maternal characteristics and pharmaceutical treatment of gestational diabetes: an analysis of the UK Born in Bradford (BiB) cohort study" (p.1 of the manuscript).

*Minor grammatical corrections need to be done in the abstract.

Response: Major revisions have been made to the abstract and grammatical errors were checked for (p.2 of the manuscript).

VERSION 2 – REVIEW

REVIEWER	Nagasubramanian, Vanitha Sri Ramachandra Institute of Higher Education and Research, Faculty of Pharmacy
REVIEW RETURNED	13-Sep-2021
GENERAL COMMENTS	Good work.